# A Novel Nondestructive Testing Probe Using AlN-Based Piezoelectric Micromachined Ultrasonic Transducers (PMUTs)

**DOI:** 10.3390/mi15030306

**Published:** 2024-02-23

**Authors:** Jiawei Yin, Zhixin Zhou, Liang Lou

**Affiliations:** 1School of Microelectronics, Shanghai University, Shanghai 201800, China; 2The Shanghai Industrial µTechnology Research Institute, Shanghai 201899, China

**Keywords:** ultrasonic nondestructive testing, piezoelectric micromachined ultrasonic transducers, time of flight, bandwidth

## Abstract

Ultrasonic nondestructive testing (NDT) usually utilizes conventional bulk piezoelectric transducers as transceivers. However, the complicated preparation and assembly process of bulk piezoelectric ceramics limits the development of NDT probes toward miniaturization and high frequency. In this paper, a 4.4 mm × 4.4 mm aluminum nitride (AlN) piezoelectric micromachined ultrasonic transducer (PMUT) array is designed, fabricated, characterized, and packaged for ultrasonic pulse–echo NDT of solids for the first time. The PMUT array is prepared based on the cavity silicon-on-insulator (CSOI) process and packaged using polyurethane (PU) material with acoustic properties similar to water. The fabricated PMUT array resonates at 2.183 MHz in air and at around 1.25 MHz after PU encapsulation. The bandwidth of the packaged PMUT receiver (244 kHz) is wider than that of a bulk piezoelectric transducer (179 kHz), which is good for axis resolution improvement. In this work, a hybrid ultrasonic NDT probe is designed using two packaged PMUT receivers and one 1.25 MHz bulk transmitter. The bulk transmitter radiates an ultrasonic wave into the sample, and the defect echo is received by two PMUT receivers. The 2D position of the defect could be figured out by time-of-flight (TOF) difference, and a 30 mm × 65 mm detection area is acquired. This work demonstrates the feasibility of applying AlN PMUTs to ultrasonic NDT of solids and paves the way toward a miniaturized NDT probe using AlN PMUT technology.

## 1. Introduction

Ultrasonic testing has been widely adopted in industrial nondestructive testing (NDT) for its excellent penetration capability, large detection area, low cost, and rapid detection. Conventional bulk piezoelectric transducers are often employed as transceivers in ultrasonic testing for their high output sound pressure. However, bulk ultrasonic probes are difficult to apply in some space-limited scenarios because of their large size. Even worse, the complex “dice-and-fill” process of conventional ultrasonic phased arrays limits the downscaling of an array pitch, which prevents the development of high-frequency applications [1].

Fortunately, micromachined ultrasonic transducers (MUTs) have significant advantages over conventional bulk piezoelectric transducers in terms of miniaturization and integration. According to their working principle, MUTs can be divided into piezoelectric micromachined ultrasonic transducers (PMUTs) and capacitive micromachined ultrasonic transducers (CMUTs). Compared with CMUTs, PMUTs have the advantages of low bias voltage, good linearity, and low energy loss.

Lead zirconium titanate (PZT) and aluminum nitride (AlN) are commonly used piezoelectric materials in PMUTs. PZT PMUTs tend to have better transmission performance due to the high piezoelectric coefficient of PZT, while AlN PMUTs feature better reception performance because of AlN’s low dielectric coefficient, even though it has a low piezoelectric coefficient. Nowadays, AlScN attracts more attention for its relatively high piezoelectric coefficient [2,3]. Moreover, both AlScN and AlN are CMOS compatible, which is more convenient in integration with circuits toward miniaturization [4]. PMUTs have many applications in the gas and fluid domains: in-air range detection [5,6], MEMS hydrophone [7], underwater communication [8], underwater wireless power supply [9], ultrasonic flowmeter [10], Doppler blood flowmeter [11], and fingerprint sensor [12,13,14]. Several studies on utilizing PMUTs for solid internal inspection have been conducted. Z. Xing et al. designed and fabricated PZT PMUTs for solid thickness measurement [15]. W. Ji et al. designed and implemented an NDT imaging system based on a PZT PMUT array [16]. Although PZT PMUTs have superior transmission performance compared with AlN PMUTs, the output sound pressure of PZT PMUTs is still lower than that of conventional bulk piezoelectric transducers. Thus, the imaging and testing area is limited, and the SNR of the defect echo is low [16]. M. Kabir et al. employed AlN PMUTs as acoustic emission (AE) sensors to monitor the structural health of solids by receiving the elastic waves excited by the generation of defects [17]. The AE NDT method can monitor the generation progress of defects in solids but cannot detect the defects that already exist. Solid inner flaws could excite the second harmonic of the incident ultrasonic wave due to the nonlinear effect, and H. Kazari et al. designed and fabricated AlN PMUTs to receive the second harmonic to detect internal defects [18]. Both AE and nonlinear acoustic NDT methods based on AlN PMUTs reported in previous work [17,18] are especially susceptible to noise interference, and the precise localization of defects is hard to achieve.

In this paper, a 4.4 mm × 4.4 mm AlN PMUT array is designed, fabricated, and characterized. The PMUT array is encapsulated as an ultrasonic testing receiver by a polyurethane (PU) material. In the pulse receiving experiment, the −3 dB bandwidth of the reported PMUT receiver is better than that of conventional piezoelectric transducers. A wide bandwidth is of great importance for improving the axis resolution of pulse–echo detection. In addition, a conventional bulk piezoelectric transducer is utilized as the transmitter to radiate ultrasonic waves into the solid sample, and the defect echo is received by PMUT receivers. Meanwhile, multiple receivers are introduced to achieve 2D defect localization by time-of-flight (TOF) difference. Thanks to the excellent transmission performance of the conventional transducer and the sensitive reception performance of PMUT, the pulse–echo NDT method reported in this paper can detect defects at a depth of 100 mm.

## 2. Design and Fabrication of AlN-Based PMUT NDT Sensor

### 2.1. PMUT Model Fabrication and Characterization

The fundamental structure of PMUTs is a laminate consisting of an upper electrode, a piezoelectric thin film, and a bottom electrode. Figure 1a depicts the structure of the PMUT fabricated in this paper, which consists of an AlN piezoelectric layer sandwiched by two Mo electrode layers, a Si passive layer, buried oxide, Al connection wires, and a SiO_2_ dielectric layer. When an AC voltage is applied to the electrodes, the composite layer vibrates periodically due to the inverse piezoelectric effect and radiates ultrasonic waves into the medium. When the incident ultrasonic wave deforms the composite layer, an electric potential difference is generated between electrodes by the piezoelectric effect.

Unlike conventional bulk piezoelectric transducers whose resonant frequency is only related to material properties and bulk thickness, PMUTs vibrate in the flexural mode, and the resonant frequency of the PMUTs depends on piezoelectric material properties, laminate thickness, and diaphragm size. For circular diaphragm PMUTs, the first-order resonant frequency can be calculated by the following Equation (1) [5]:(1)$$f_{r}=\frac{0.47t\sqrt{E^{\prime }/\rho_{m}}}{r^{2}}$$
where *t* is the total thickness of the composite laminate, *E*′ is the average plate Young’s modulus, *ρ_m_* is the average density of the vibrating diaphragm, and *r* is the radius of the PMUT cavity. The resonant frequency of the transmitter used in this work was 1.25 MHz, and the resonant frequency of PMUTs should be around 1.25 MHz after PU encapsulation to obtain excellent receiving sensitivity. Finite element method (FEM) simulations were performed in COMSOL Multiphysics to find the appropriate geometric parameters of the PMUT with the desired performance. The PU encapsulation material has similar properties to water, so we chose water as the package material in COMSOL simulations. Based on SITRI’s fabrication experience, a 1 μm AlN piezoelectric layer, 0.2 μm Mo electrodes, and a passive layer consisting of 5 µm Si and 1 μm SiO_2_ were adopted to compromise fabrication capability and device performance. With the thickness of the AlN piezoelectric layer increasing, the etching process becomes more and more difficult, and the problem of incomplete etching was posed accordingly. For a fixed material and thickness of each layer, the resonant frequency of the PMUTs is determined by the cavity size. Considering the circle diaphragm, a 2D rotational symmetry model was employed for simulation optimization.

The simulation results are shown in Figure 1c, the inset is the simulated first-order resonant mode. The resonant frequency of the PMUT in liquid decreases due to the mass loading effect introduced by the liquid medium [19]. According to the resonant frequency–radius spectrums plotted in Figure 1c, a PMUT with a 110 μm radius circular diaphragm resonates at 2.25 MHz in air and at 1.25 MHz when placed in water. Based on the simulation results, the radius of the cavity was selected as 110 μm. The geometry parameters of the designed PMUT are listed in Table 1. Furthermore, previous research indicates that PMUTs could achieve optimal performance when the top electrode coverage is 70% [1]. Additionally, in order to achieve better receiving sensitivity, an 18 × 18 PMUT array was designed with all elements parallel connected.

The fabrication process flow of the PMUT array is shown in Figure 2. The process flow started from a custom CSOI (cavity silicon-on-insulator) wafer with a 5 μm silicon passive layer and 1 μm buried oxide as shown in Figure 2a. First, a 50 nm AlN seed layer was deposited. Then, 0.2/1/0.2 μm Mo/AlN/Mo were deposited by physical vapor deposition (PVD) under low temperature (<400 °C) in Figure 2b. And then, the top Mo electrode was patterned by reactive ion etching (RIE) for the designed top electrode coverage in Figure 2c. Afterward, the SiO_2_ dielectric layer was deposited by plasma-enhanced chemical vapor deposition (PECVD), and the electrical vias were defined by RIE as in Figure 2d. Subsequently, the Al layer was deposited by PVD and patterned by dry etching to form reliable electrical connecting wires as in Figure 2e. At the end of the process flow, the SiO_2_/AlN/Mo/Si stack was etched to achieve 5 μm wide isolation trenches as in Figure 2f. By etching isolation trenches, the residual stress within the piezoelectric layer induced by fabrication processes could be released [20], and the PMUT performance would be optimized. The optical image of the fabricated PMUT array is shown in Figure 2g.

The electrical characteristics of the fabricated PMUT array were measured by an impedance analyzer (E4990A, Keysight Technologies, Santa Rosa, CA, USA). The impedance properties of the PMUT array in air are depicted in Figure 3a. According to the IEEE Standard [21], the effective electromechanical coupling coefficient (*k_eff_*^2^) of the PMUT array could be calculated by the following Equation (2):(2)$${k_{eff}}^{2}=\frac{{f_{a}}^{2}-{f_{r}}^{2}}{{f_{a}}^{2}}$$
where *f_r_* is the resonant frequency, and *f_a_* is the anti-resonant frequency. From the impedance–frequency spectrum, the resonant frequency of the PMUT array was 2.183 MHz, and the anti-resonant frequency was 2.203 MHz. The calculated effective electromechanical coefficient was 1.8%. The electromechanical properties of PMUTs reported in previous works are listed in Table 2 for comparison, which shows that the PMUT array reported in this paper had great electromechanical performance.

The mechanical vibration properties were characterized by a laser Doppler vibrometer (MSA-600, Polytec GmbH, Waldronn, Germany). The displacement–frequency spectrums of the same PMUT element in air and deionized water are depicted in Figure 3b,c, and the insets are the first resonant mode of the PMUT. In air, the first-order resonant frequency of the PMUT was 2.2925 MHz, and the −3 dB bandwidth was 5.3 kHz. When immersed in deionized water, the resonant frequency of the PMUT decreased to 1.2206 MHz, and the −3 dB bandwidth expanded to 33.84 kHz, because of the mass loading effect [19]. The in-water −3 dB bandwidth of the PMUTs prepared in this paper is larger than that (20.3 kHz) in previous work [10]. The quality factor (*Q*) of the PMUTs is defined by the following Equation (3):(3)$$Q=\frac{f_{r}}{BW_{-3dB}}$$

The *Q* in air was 432, and it dropped to 36 in deionized water. According to the characterization results, the PMUT array prepared in this paper had low *Q* and a wide −3 dB bandwidth when working in water. The acoustic properties of PU encapsulation material are similar to water, and the packaged PMUT sensor also had a wide bandwidth.

### 2.2. PMUT Package and 2D Defect Localization

The PMUT array chips were first fixed to the printed circuit boards (PCBs) using an adhesive. The electrical connections between the PMUT chip and the PCB were conducted through wire bonding. Afterward, the PCB with the PMUT array chip was packaged in an acrylic plastic housing and covered by a protecting layer. To better protect the PMUT arrays and achieve good coupling between the sample and the PMUT chip, a polyurethane (PU) material was employed as an encapsulation and acoustic impedance matching medium. The acoustic impedance of PU is about 1.57 MRayl, which is close to that of water and common NDT coupling agents. One conventional bulk piezoelectric transducer (as a transmitter) and two packaged PMUT receivers were assembled in one acrylic plastic housing to produce a hybrid NDT probe. The structure of the hybrid ultrasonic NDT probe is shown in Figure 4a, and Figure 4b is the optical image of the assembled probe. The geometry size of the produced hybrid probe was 42 mm × 14 mm × 10 mm, which is smaller than conventional bulk double-crystal probes.

The position of the defect could be determined by the TOF difference between two different receivers. Figure 4c illustrates the theory of 2D defect localization. The ultrasonic wave radiated into the sample by the bulk piezoelectric transducer will be reflected by the inner defect and received by the PMUT receivers. The TOF of Receiver A and Receiver B is *t_A_* and *t_B_*, and the longitudinal sound velocity in the sample is *c* (*c* ≈ 6000 m/s); we then have the following:(4)$$l_{1}=c\times t_{A}$$
(5)$$l_{2}=c\times t_{B}$$
(6)$$x^{2}+y^{2}=d^{2}$$
(7)$${\left(x-a\right)}^{2}+y^{2}={\left(l_{2}-d\right)}^{2}$$
(8)$${\left(x+a\right)}^{2}+y^{2}={\left(l_{1}-d\right)}^{2}$$
where *l*_1_ and *l*_2_ are the propagation distance of the ultrasonic wave between the transmitter and the receiver, and *a* is the horizontal distance between the transmitter and the receiver. Circles with the positions of Receiver A and Receiver B as the centers, with *l*_1_
*− d* and *l*_2_
*− d* as radiuses, have two intersections. And *d* could be derived from the Equations (4)–(8) above:(9)$$d=\frac{{l_{1}}^{2}+{l_{2}}^{2}-2a^{2}}{2(l_{1}+l_{2})}$$

And then, the intersections of these two circles could be figured out. The intersection of the two circles under the *X*-axis is the location of the defect.

### 2.3. Ultrasonic Testing Simulation

The propagation progress of the ultrasonic wave from the transmitter to the receiver was modeled and simulated in COMSOL Multiphysics 6.0. The model was divided into two parts: one for an ultrasonic wave propagating in the solid, being reflected by the defect and transmitting into the matching layer, and the other for an ultrasonic wave being received by the PMUT from the matching layer.

The first simulation model (Acoustic–Solid Interaction, Time Explicit) is depicted in Figure 5a. The sample was a 100 mm × 100 mm aluminum square containing one circular hole with 2 mm diameter at a depth of 50 mm. The transmitter was a plastic block to simulate the cover of the conventional piezoelectric probe. The encapsulation layer of the PMUT was modeled as a water matching layer surrounded by an absorbing layer. A Gaussian-pulse normal velocity was applied at the surface of the sample to simulate the ultrasonic wave radiated by the transmitter. Figure 5b illustrates the pressure distribution in the solid sample when the ultrasonic wave is reflected by the defect. And then, the ultrasonic pulse will be reflected by the defect and incident into the matching layer. The sound pressure in the matching layer was acquired by a COMSOL probe.

The second simulation model (Pressure Acoustics, Time Explicit) is shown in Figure 5c. The PMUT was placed in a semicircle water domain covered by a perfect matching layer. A line source with sound pressure acquired in the first part was placed 1 mm above the PMUT. The PMUT was modeled as a laminate consisting of Mo electrodes, an AlN piezoelectric layer, a Si vibrating layer, and buried SiO_2_. Figure 5d presents the distribution of sound pressure in the water medium. The normal excitation velocity and received signals of defects located at 50/75/100 mm depth are depicted in Figure 5c. The TOF of the 50/75/100 mm defect was 18.27/26.14/32.36 μs.

## 3. Results and Discussion

### 3.1. Ultrasonic Testing Experiment

The setup of the ultrasonic testing experiment and test sample are shown in Figure 6. A conventional bulk piezoelectric transducer with a 1.25 MHz resonant frequency was employed as the transmitter. A three-cycle sine wave with 1.25 MHz frequency and 6 Vpp amplitude was generated by a function generator (33600A, Keysight Technologies, Santa Rosa, CA, USA). Then, the burst signal was amplified 25 times by a voltage amplifier (WMA-300, Falco Systems, Katwijk aan Zee, The Netherlands) to drive the transmitter. Furthermore, 6061 aluminum alloy cubes with 2 mm diameter holes located 50/73/97 mm deep were prepared as test samples. The output signals of the PMUT receivers were amplified and denoised by a front analog processing circuit (30 dB, −3 dB width, 272 kHz–2.2 MHz). Afterward, the received signals were displayed and acquired by an oscilloscope (DSOX2014A, Keysight Technologies, Santa Rosa, CA, USA). A gel ultrasonic matching agent was coated between the sensors and test samples for coupling.

The acquired signal of a 50 mm deep defect directly under the transmitter is depicted in Figure 7a. In the experiment, the transmitter was not an ideal ultrasonic wave source, and the emission crosstalk became much more severe than that simulated by COMSOL Multiphysics. Although the received signal was denoised by the front analog bandpass circuit, there was still much noise that could not be canceled by traditional denoising methods. As a result, the SNR of the defect echo was poor (13.14 dB), and the accuracy and resolution of the detection were affected.

Wavelet denoising is a suitable conditioning method for ultrasonic echo signals [26]. Based on the wavelet transform theory, the original signal was decomposed into many wavelet bases, and noise signals with features different form the ultrasonic pulse were eliminated. The ultrasonic pulse was kept and reconstructed as a denoised signal. The acquired signal was postprocessed with wavelet denoising using MATLAB’s Wavelet Signal Denoiser tool (R2021a). The wavelet denoised signal is also depicted in Figure 7a. After wavelet denoising, the SNR of defect echo was 33.52 dB, which is greatly improved compared with the original signal. From the wavelet denoised signal, the TOF of the defect echo was 16.64 μs whereas the simulated TOF was 18.27 μs, which is caused by the model deviation. When the transmitter is not directly above the defect, the TOFs of the defect echo received by Receiver A and Receiver B are different. The relative position of the defect can be figured out by the difference in the TOFs. The received signals of defects located at different relative positions are shown in Figure 7b–d.

For the 50 mm deep defect, the hybrid NDT probe was moved horizontally across 5 mm, and the signal received by Receiver A and Receiver B is depicted in Figure 7b. The defect echo received by Receiver A arrived at 16.53 μs with an amplitude of 78 mVpp, and the defect echo received by Receiver B arrived at 17.02 μs with an amplitude of 57 mVpp. The vertical distance between the transmitter and the defect was calculated to be 49.07 mm, and the horizontal distance was 5.38 mm. For the 73 mm deep defect located 10 mm to the left of the transmitter, *t_A_* was 24.30 μs and *t_B_* was 24.88 μs, and the amplitude of the defect echo received by Receiver A and Receiver B was 76 mVpp and 68 mVpp, as depicted in Figure 7c. The vertical distance was calculated to be 72.53 mm, and the horizontal distance was 9.25 mm. For the 97 mm deep defect located 15 mm to the left of the transmitter, *t_A_* was 32.12 μs and *t_B_* was 32.82 μs; the amplitude of the defect echo received by Receiver A and Receiver B was 72 mVpp and 51 mVpp, as shown in Figure 7d. The vertical distance was calculated to be 95.50 mm, and the horizontal distance was 14.69 mm. With the vertical and horizontal distance increasing, the TOF differences between Receiver A and Receiver B become more obvious. Compared with the simulated signal, severe electromagnetic feedback was found during 0–10 μs, which could cause about a 30 mm blind zone.

The horizontal detection range of the reported NDT probe is limited by the sound beam width of the transmitter and the ultrasonic wave attenuation of propagation and reflection. When the horizontal shift distance exceeds 15 mm, the outer sensor can barely receive the defect echo, and the horizontal range of the NDT probe prepared in this paper is around 30 mm.

### 3.2. Pulse Receiving Experiment

Pulse receiving experiments are performed to study the receiving performance of the PMUT receiver. The schematic of the receiving experiments is illustrated in Figure 8a. One conventional bulk piezoelectric transducer with 1.25 MHz resonant frequency was employed as the transmitter. In the experiment, the transmitter and receiver were placed on two sides of a 4 cm thick 6061 aluminum alloy cube. The transmitter was excited by a five-cycle sine wave signal with an amplitude of 150 Vpp amplified by a voltage amplifier. The received signals were acquired by the oscilloscope and are shown in Figure 8c. The pulse signal received by the PMUTs was amplified and denoised by a preconditioning circuit. A conventional piezoelectric probe without conditioning circuits was also employed as the receiver for comparison. The sensors and the alloy cube were coupled by an ultrasonic matching agent.

From the time domain, the pulse signal received by the PMUT sensor arrived at 7.19 μs, and the pulse signal received by the conventional bulk probe arrived at 6.86 μs due to the difference in thickness between the matching layers. The amplitude of the defect echo received by the PMUT sensor was 1.29 Vpp and that of the conventional bulk probe was 0.83 Vpp. Furthermore, better reception sensitivity will be acquired by using AlScN PMUTs as receivers because of AlScN’s higher piezoelectric coefficient [2,3]. For the PMUT receiver, the ultrasonic pulse is reflected by the PMUT chip and PCB, and the reflected wave transmits to the interface of the matching layer and sample. Then, the ultrasonic pulse is reflected by the sample, and a second echo is generated. 

Moreover, a fast Fourier transform (FFT) of the received pulse was conducted to analyze the frequency components. The normalized amplitude–frequency spectrum is plotted in Figure 8d. The −3 dB bandwidth of the PMUT receiver (244 kHz) was 36.3% wider than that of the conventional bulk transducer (179 kHz). The PMUT receiver had better bandwidth, and the spectrum concentrated around 1.25 MHz, whereas the signal received by the bulk piezoelectric probe had some extra high-frequency noise. Bulk piezoelectric probes are often designed with additional damping blocks to achieve high damping and large bandwidth; however, the PMUT receiver prepared in this paper is packaged without any damping blocks. The bandwidth of the PMUT receiver fabricated in this paper was significantly better than the PMUT sensor prepared for AE detection in previous work [17]. In all, the PMUT receiver reported in this paper had better bandwidth and tiny size compared with a conventional NDT bulk piezoelectric transducer, which is essential for the miniaturization and integration of high-axis-resolution NDT detection systems.

## 4. Conclusions

In this paper, a 1.25 MHz hybrid ultrasonic pulse–echo NDT probe employing AlN PMUT arrays as receivers and a conventional bulk transducer as the transmitter is initiatively reported. The PMUT’s electromechanical characterization and pulse receiving experiments indicate that the packaged PMUT receiver has a wider bandwidth (244 kHz) than a bulk piezoelectric transducer (179 kHz), which is crucial for axis resolution improvement in ultrasonic testing. The 2D position of the defect is detected through TOF difference, and a 30 mm × 65 mm detection area is acquired. This work demonstrates the feasibility of applying AlN PMUTs to ultrasonic nondestructive evaluation and will promote the miniaturization and integration of ultrasonic NDT systems. In the future, employing AlScN PMUTs as transceivers will further improve the sensitivity and integration of ultrasonic NDT systems for their better performance.

## Figures and Tables

**Figure 1 micromachines-15-00306-f001:**
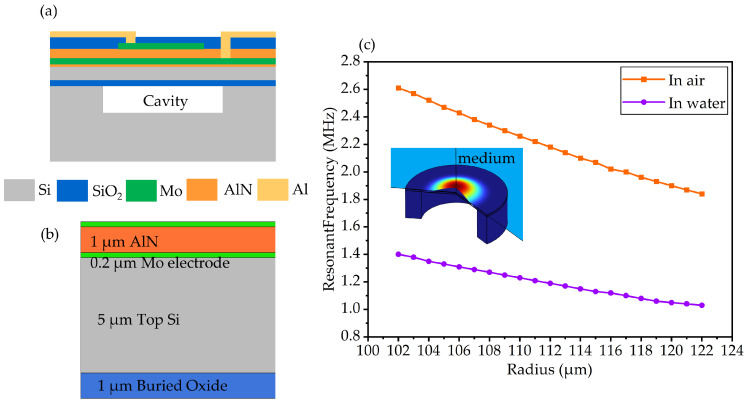
(**a**) Structure of the PMUT. (**b**) Cross-sectional view of the FEM simulation model. (**c**) Simulation results of first-order resonant frequency in air and water with different radiuses.

**Figure 2 micromachines-15-00306-f002:**
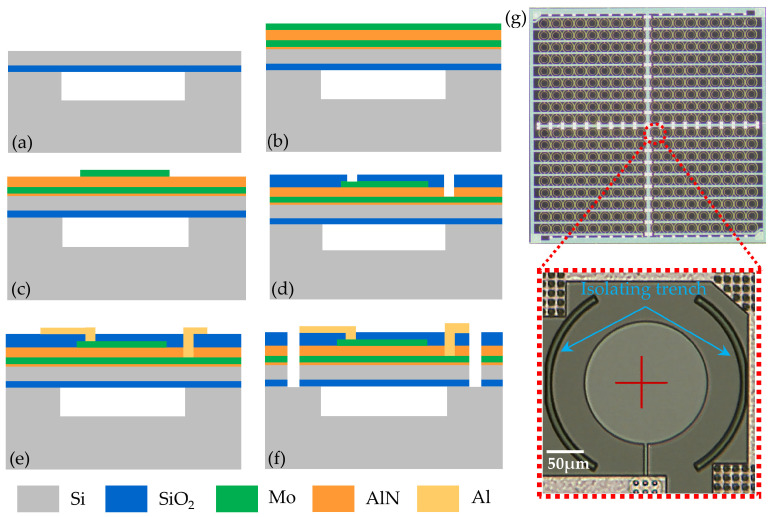
(**a**–**f**) Fabrication process flow of the PMUT array. (**g**) Optical image of the fabricated PMUT array.

**Figure 3 micromachines-15-00306-f003:**
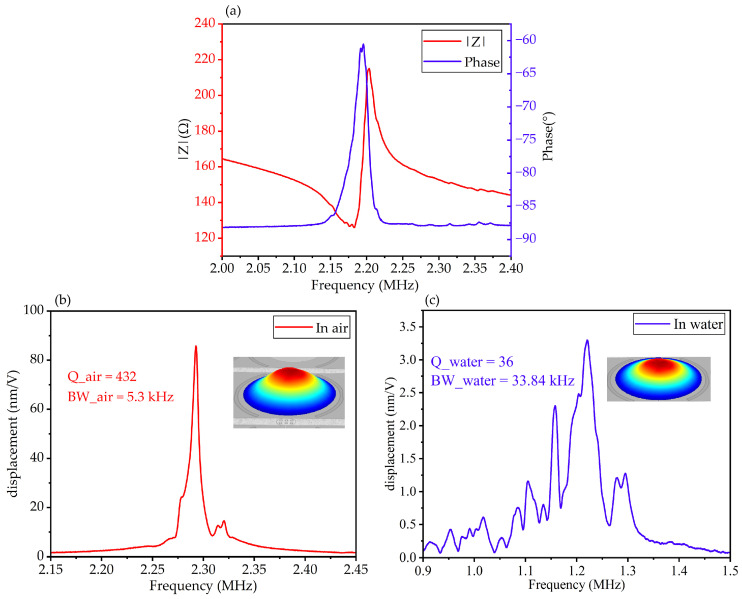
Electromechanical characterization of the PMUT array. (**a**) Measured electrical impedance in air. (**b**) Measured mechanical vibration performance in air. (**c**) Measured mechanical vibration performance in deionized water.

**Figure 4 micromachines-15-00306-f004:**
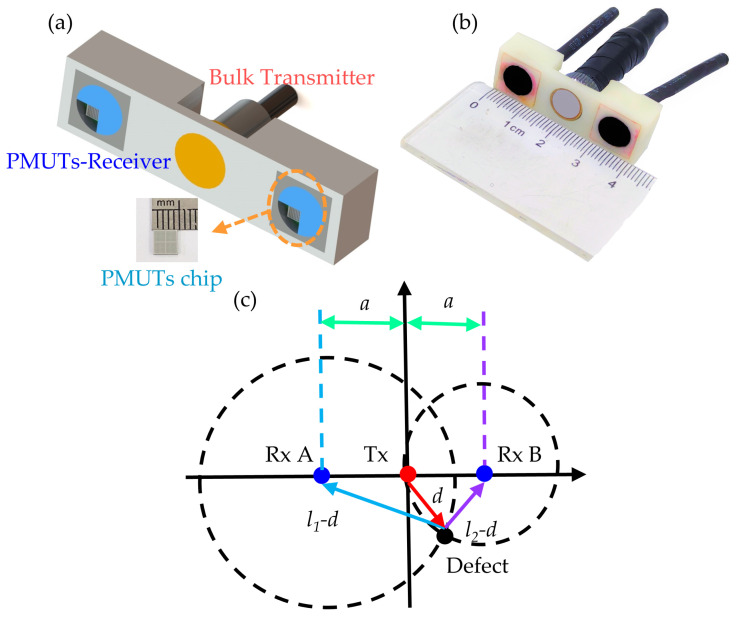
(**a**) Structure of the hybrid NDT probe. (**b**) Optical image of the NDT probe. (**c**) Schematic of 2D defect localization.

**Figure 5 micromachines-15-00306-f005:**
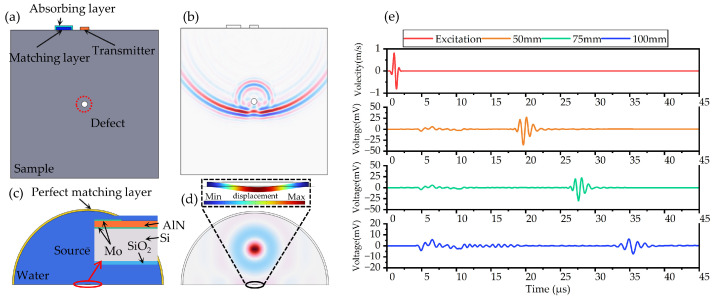
Propagation simulation. (**a**) Schematic of the first simulation model. (**b**) Pressure distribution in solid sample. (**c**) Schematic of the second simulation model. (**d**) Sound pressure distribution in matching layer. (**e**) Excitation and defect echo from different depths.

**Figure 6 micromachines-15-00306-f006:**
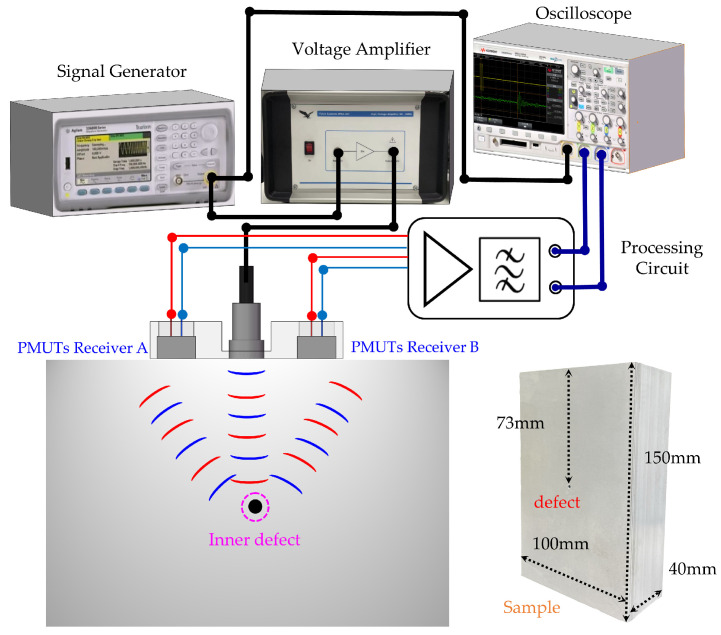
Setup of the ultrasonic testing experiment.

**Figure 7 micromachines-15-00306-f007:**
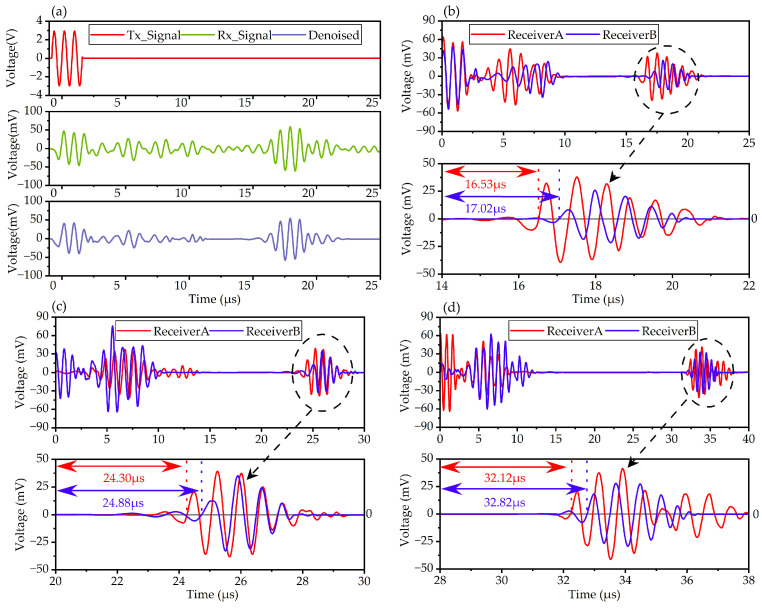
Defect echo received by PMUT sensor: (**a**) 50 mm deep defect echo, (**b**) 50 mm deep and 5 mm horizontal shift defect echo, (**c**) 73 mm deep and 10 mm horizontal shift defect echo, (**d**) 97 mm deep and 15 mm horizontal shift defect echo.

**Figure 8 micromachines-15-00306-f008:**
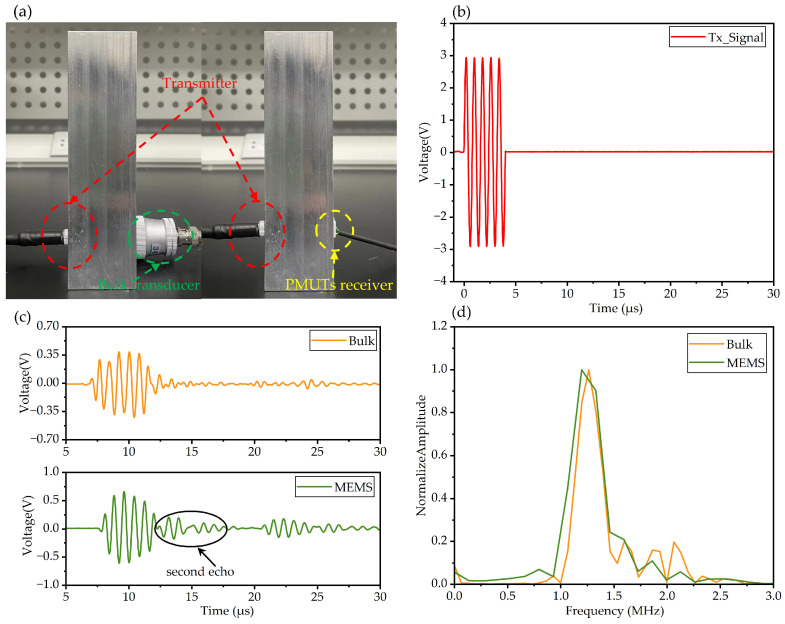
(**a**) Schematic of the receiving experiment. (**b**) Excitation signal. (**c**) Received signal by bulk transducer and MEMS sensor. (**d**) FFT of the received signal.

**Table 1 micromachines-15-00306-t001:** Geometric parameters of designed PMUT.

Material	Top Mo	AlN	Bottom Mo	Si	SiO_2_	Cavity
Radius (μm)	77	-	-	-	-	110
Thickness (μm)	0.2	1	0.2	5	1	-

**Table 2 micromachines-15-00306-t002:** Comparison of effective electromechanical coupling coefficient between PMUT array prepared in this paper and previous work.

Author	Piezoelectric Layer	Chip Size	Resonant Frequency	*k_eff_* ^2^
J. Ling et al. [22]	1 μm PZT	15 mm × 7.5 mm	0.753 MHz	1.82%
K. Zhu et al. [23]	1 μm AlN	-	0.484 MHz	1.54%
E. Ledesma et al. [24]	0.6 μm AlN	-	4.866 MHz	1.14%
Y. Gao et al. [10]	1 μm AlSc_10_N	2.8 mm × 2.8 mm	1.996 MHz	1.38%
Z. Shao et al. [25]	Bimorph 1 μm AlN	-	133 kHz	3.03%
Q. Wang et al. [2]	1 μm AlSc_15_N	-	17 MHz	1.9%
This work	1 μm AlN	4.4 mm × 4.4 mm	2.183 MHz	1.8%

## Data Availability

Data are contained within the article.

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
