# Peer review of "A Novel Nondestructive Testing Probe Using AlN-Based Piezoelectric Micromachined Ultrasonic Transducers (PMUTs)"

_micromachines, 2024, doi:10.3390/mi15030306_

Round 1

Reviewer 1 Report

Comments and Suggestions for Authors

The article focuses on a new method for non-destructive testing of received ultrasonic pulse echoes using AlN-based array PMUTs. The article describes in detail the design, fabrication, and characterization of PMUTs and discusses their application in the detection of solid samples. The authors point out that PMUTs have a wider bandwidth and smaller size than conventional piezoelectric transducers, which are suitable for miniaturization and integration of NDT systems that achieve high axial resolution. Through experimental results, the authors demonstrate the feasibility of AlN PMUTs for ultrasonic pulse-echo NDT and highlight their potential applications in the detection and localization of internal defects in solid materials.

This paper focuses on simulation design, PMUT fabrication, and performance verification. There is a more coherent logic, but it lacks a lot of theoretical analysis and data processing. Among them, the data processing of experimental results is more lacking, and it is recommended to focus on supplementing!

It is recommended to optimize the zoomed-in diagram of the array element in Fig. 2. As well as the poor clarity of the text in the figure.

It is recommended to optimize the connecting part in Fig. 6.

In Section 3 only a brief description of the signal is given, and it is recommended to add quantitative comparisons:

In Section 3.1, lines 280-289, only the data are described, with no relevant comparison discussion. There is also no mention of the connection between the previous simulation and this experiment. It is suggested that the authors add it.

Comments on the Quality of English Language

The language presentation of this paper needs to be strengthened, and it is suggested that the authors focus on optimizing and embellishing the English presentation of the first part.

Author Response

The authors appreciate the valuable comments from the reviewers and have strived to improve the manuscript according to the provided suggestions. Please see our response in the attached file.

Reviewer 2 Report

Comments and Suggestions for Authors

The authors present a hybrid ultrasonic NDT probe based on PMUT and bulk piezoelectric transducer. The results show the feasibility of applying PMUT to ultrasonic NDT probes with a better receive sensitivity. The work is well-written and falls within this journal’s scope. However, there are still some comments to be addressed.

1.       In Figure 2(g), please add scale bars.

2.       In Table 2, the authors should include the following references for a more comprehensive comparison of electromechanical coupling coefficient:

Shao, Z, et al. "Bimorph pinned piezoelectric micromachined ultrasonic transducers for space imaging applications." Journal of Microelectromechanical Systems 30.4 (2021): 650-658.

Wang, Q, et al. "Design, fabrication, and characterization of scandium aluminum nitride-based piezoelectric micromachined ultrasonic transducers." Journal of microelectromechanical systems 26.5 (2017): 1132-1139.

3.       Please also measure and characterize the bulk piezoelectric transducer, similar to characterizing the PMUT, e.g. impedance, frequency spectrum, center frequency, bandwidth, etc.

4.       In part 2.3, please explain why two separate models are needed. Why not just use a single model to combine the piezoelectric transducer and PMUT? And for PMUT simulation, the authors should include structural mechanics in CMOSOL to capture the PMUT membrane vibration. And please add a plot of simulated maximum PMUT membrane displacement caused by the received ultrasound waves.

5.       In part 3.1, please explain what the principle of wavelet denoising is and how it works.

6.       In part 3.1, the authors should reconstruct the ultrasound NDT image based on the acquired ultrasound signal. And please analyze the axial and lateral resolution after reconstructing the image.

Author Response

(The authors gave the same response as above.)

Round 2

Reviewer 1 Report

Comments and Suggestions for Authors

After the revision, the language has been improved.

Reviewer 2 Report

Comments and Suggestions for Authors

The authors should put more effort into imaging and characterization experiments.